# Absence of Stress Hyperglycemia Indicates the Most Severe Form of Blunt Liver Trauma

**DOI:** 10.3390/diagnostics11091667

**Published:** 2021-09-13

**Authors:** Janett Kreutziger, Margot Fodor, Dagmar Morell-Hofert, Florian Primavesi, Stefan Stättner, Eva-Maria Gassner, Stefan Schmid, Christopher Rugg

**Affiliations:** 1Department of Anesthesiology and Critical Care Medicine, Medical University of Innsbruck, 6020 Innsbruck, Austria; stefan.schmid@tirol-kliniken.at (S.S.); christopher.rugg@tirol-kliniken.at (C.R.); 2Department of Visceral, Transplant and Thoracic Surgery, Medical University of Innsbruck, 6020 Innsbruck, Austria; margot.fodor@i-med.ac.at; 3Department of Radiology, Medical University of Innsbruck, 6020 Innsbruck, Austria; dagmar.morell@i-med.ac.at (D.M.-H.); eva.gassner@tirol-kliniken.at (E.-M.G.); 4Department of General, Visceral and Vascular Surgery, Salzkammergut Klinikum Vöcklabruck, 4840 Vöcklabruck, Austria; florian.primavesi@ooeg.at (F.P.); s.staettner@icloud.com (S.S.)

**Keywords:** stress hyperglycemia, liver injury, kidney injury, spleen injury, trauma, outcome, AAST, blood glucose, diagnostic criterion, parenchymatous organs

## Abstract

Background: Stress hyperglycemia is common in trauma patients. Increasing injury severity and hemorrhage trigger hepatic gluconeogenesis, glycogenolysis, peripheral and hepatic insulin resistance. Consequently, we expect glucose levels to rise with injury severity in liver, kidney and spleen injuries. In contrast, we hypothesized that in the most severe form of blunt liver injury, stress hyperglycemia may be absent despite critical injury and hemorrhage. Methods: All patients with documented liver, kidney or spleen injuries, treated at a university hospital between 2000 and 2020 were charted. Demographic, laboratory, radiological, surgical and other data were analyzed. Results: A total of 772 patients were included. In liver (*n* = 456), spleen (*n* = 375) and kidney (*n* = 152) trauma, an increase in injury severity past moderate to severe (according to the American Association for the Surgery of Trauma, AAS**T** III-IV) was associated with a concomitant rise in blood glucose levels independent of the affected organ. While stress-induced hyperglycemia was even more pronounced in the most severe forms (AAST V) of spleen (median 10.7 mmol/L, *p* < 0.0001) and kidney injuries (median 10.6 mmol/L, *p* = 0.004), it was absent in AAST V liver injuries, where median blood glucose level even fell (5.6 mmol/L, *p* < 0.0001). Conclusions: Absence of stress hyperglycemia on hospital admission could be a sign of most severe liver injury (AAST V). Blood glucose should be considered an additional diagnostic criterion for grading liver injury.

## 1. Introduction

Stress hyperglycemia is common in trauma patients, and critical illness on hospital admission and is often associated with poor outcome [1,2,3,4,5,6,7]. This stress hyperglycemia is caused by neuroendocrine, inflammatory and metabolic responses to trauma-associated stressors. Pain, anxiety and psychogenic stress may lead to activation of the sympathetic nervous system and the hypothalamic–pituitary axis, increasing circulating catecholamines. A traumatic brain injury may even lead to a sympathetic storm [8,9], thus further amplifying this neuroendocrine response. Another very strong trigger for a stress response such as stress hyperglycemia is hemorrhage [10], which is mediated by baroreceptors and multifactorially released cytokines [11]. Furthermore, tissue hypoxemia and injury, endothelial damage and complement activation [12,13] kick off an avalanche of immune-stimulating and -modulating chemokines, interleukins [14,15,16], damage-associated molecular patterns (DAMPS) and multiple cell line activations [13,17]. Circulating and locally secreted tumor necrosis factor (TNF)α [18,19] and catecholamines [20] lead to hepatic insulin resistance, which annuls glucose homeostasis. By triggering rampant hepatic gluconeogenesis and glycogenolysis both processes are decoupled from physiological feedback mechanisms [21,22,23] and, additionally, are fed by free fatty acids released by catecholamines from fat tissue, [24,25] but also from lactate and alanine of muscle tissue [25,26,27].

TNFα-associated hepatic insulin resistance is probably mediated by TNF receptors by activating a kinase (c-Jun-NH_2_-terminal kinase), which in turn catalyzes serine phosphorylation of insulin receptor substrate 1 (IRS-1). This blocks the phosphatidylinositol 3-kinase (PI3K) and protein kinase B (AKT) pathway of intracellular metabolic effects of insulin [28]. Catecholamine (epinephrine)-mediated insulin resistance is conveyed via 3′–5′-cyclic adenosine monophosphate (cyclic AMP or cAMP) [29], activating protein kinase A (PKA) [30]. On the one hand, phosphorylase kinase is activated, itself stimulating glycogen phosphorylase and therefore leading to a disintegration of glycogen stores and release of glucose [31,32]. On the other hand, PKA inhibits in parallel substrate flux through phosphofructokinase-1 (PFK-1) (rate-limiting enzyme in glycolysis), thereby enhancing the delivery of glucose by the liver [33,34]. Moreover, peripheral insulin resistance—mainly of the muscles and probably by comparable mechanisms—limits glucose uptake and metabolism [21,35].

Probably, the pathophysiological purpose of all these varying responses to trauma is to shift energy substrates to vital organs, initiate immune defense, and repair mechanisms [20,21,22,25,26,36], which can therefore be seen as survival responses. Consequently, according to the literature, hypoglycemia is a rare finding in trauma patients at hospital admission and is triggered mainly by non-traumatic causes, such as anti-diabetic drug overdose, alcohol intoxication or chronic liver disease [37,38]. To date, there has been no consistency in defining blood glucose levels, for which reason this trial refrains from specifying any threshold values for stress hyperglycemia.

Injury severity of parenchymatous organs, such as liver, kidney or spleen, is often radiologically categorized according to the classification of the American Association for the Surgery of Trauma, AAST [39,40], Table 1. In doing so, the extent of lacerations, contusions or hematomas must be exactly measured. However, specifying proportions of parenchymal disruption of one or both hepatic lobes to distinguish between liver injury AAST IV and V may be very challenging. Active bleeding is included in AAST ≥ III injuries of the liver and kidney and AAST ≥ IV injuries of the spleen [39,40]. AAST V injuries of the liver are furthermore defined as major juxtahepatic venous injury (vena cava, central major hepatic veins) or lacerations resulting in parenchymal disruption of more than 75% of one hepatic lobe [39,40]. Assuming the liver is the primary origin of stress hyperglycemia, it seems clear that such destructive liver injuries, partially even resulting in devascularization (inflow or outflow), can impede hepatic glucose liberation provoked by trauma or hemorrhage.

Consequently, it was hypothesized that parallel to injury severity and extent of hemorrhage in hepatic, renal or splenic injuries, blood glucose levels should increase and therefore lead to significant stress hyperglycemia. However, it was also hypothesized that in the case of most severe liver injuries (AAST V), defined as major devascularization (inflow or outflow) and/or parenchymal disruption of more than 75% of one hepatic lobe, hepatic gluconeogenesis and glycogenolysis may become insufficient, consequently leading to absence of stress hyperglycemia. This may be a leading diagnostic mark.

## 2. Materials and Methods

The study was reviewed and approved by the Ethics Committee of the Medical University of Innsbruck (EK 1394/2020), and written informed consent was waived due to the observational character of the study.

This single-center analysis of detailed medical information on hepatic, splenic and renal injuries was performed by searching the local hospital information system (Krankenhausinformationssystem i.s.h.med PowerChart by Cerner Österreich GmbH, Vienna, Austria), which is prospectively maintained and auditable. A pre-existing register developed over a 17-year period (2000–2016) for the analysis of treatment strategies and associated outcomes in blunt liver, spleen and kidney injuries [41,42] was expanded up to 2020, extended to include blood glucose values on hospital admission and re-analyzed with different inclusion and exclusion criteria (Figure 1).

All trauma patients with suspected blunt hepatic, splenic or renal injury, treated at Innsbruck Medical University Hospital between 1 January 2000 and 31 August 2020 (inclusion criteria) were analyzed regarding stress hyperglycemia as a function of injury severity.

Beside the above-mentioned inclusion criteria, the following exclusion criteria were applied: missing blood glucose levels within the first hour following hospital admission, insufficient documentation of radiologic findings or laboratory analyses regarding the scientific question, admission more than 12 h beyond trauma, death on arrival and hemorrhages due to innate malformations (Figure 1).

Patients transferred from district hospitals to our Level 1 trauma center after receiving diagnostics only and within the 12-h timeframe were also included. Patients with diabetes mellitus were included. All patients underwent either selective abdominal/thoracic/cranial or whole body dual or triple phase computed tomography (CT) scans. CT images were re-evaluated for trial purposes regarding injury grading by two trained radiologists using our picture archiving and communication system (AGFA IMPAX; AGFA Health Care, Greenville, SC, USA).

Injury scoring was performed on the basis of the AAST classification for blunt liver, spleen or kidney trauma in all patients who suffered injuries to at least one of these organs and, additionally, in patients who suffered injuries to one of these organs but excluding severe co-injuries classified as Abbreviated Injury Scale (AIS) 4–6 [43] in any other organ or organ system. This was done to demonstrate the independent effects of particular organ injuries on blood glucose levels.

Primary outcome parameter was the admission blood glucose level depending on severity of injury (AAST I to V) to liver, spleen, and kidney. Secondary outcome parameters were admission blood glucose levels additionally depending on the existence of diabetes mellitus and severe co-injuries (AIS 4–6).

Descriptive statistical analysis was performed and reported proportions (%) and medians with range due to non-normal distribution of data (Shapiro–Wilk test). Overall distribution of blood glucose within the subgroups with and without co-injuries (AIS 4–6) as well as with and without diabetes mellitus was analyzed with the Kruskal–Wallis test. Within-subgroup differences (e.g., between AAST IV and V) were analyzed with the Mann–Whitney *U* test. Because of the small subgroups, two-tailed *p* values lower than 0.05 were accepted as significance level throughout the study. The confidence intervals were defined as 95% CI, accordingly. Data analysis was performed with SPSS 26.0 (IBM Corporation, Armonk, NY, USA).

## 3. Results

### 3.1. Study Population

During the study period, 879 patients presenting at our Level 1 trauma center with suspected hepatic, splenic or renal injury were assessed for eligibility. Following application of inclusion and exclusion criteria 772 patients, comprising 456 liver, 375 spleen and 152 kidney injuries, were ultimately included in the study; 189 patients had more than one organ injured (Figure 1). Median age was 29 (1–89) years, 238 (31%) patients were female. During hospitalization, 26 (3.4%) patients deceased (Table 2).

Compared to the 772 included patients, the 107 excluded patients had a comparable median age (31 (interquartile range: 16–43) years) and gender distribution (31.8% female). Of them, 50 (46.8%) suffered from hepatic, 45 (42.1%) from spleen, and six (6.6%) patients from renal injury. In sum, 13 (12.1%) patients deceased during in-hospital stay. As depicted in Figure 1, nine patients were excluded due to malformation, missing confirmation of the suspected injury or no trauma. In sum, aside from an increased—mainly early—mortality rate, the excluded population is comparable to the included population. Therefore, it was concluded that the included population is representative and inclusion of the excluded population would not have a significant impact on the presented results. Pancreatic injuries were rare in this population. Merely eight patients suffered from concomitant pancreatic injury: five pancreatic contusions were treated conservatively and three pancreatic ruptures were in need of surgical intervention. One segment resection, one surgical closure of the rupture and one complete pancreas resection were performed. Severe pancreatic involvement was mainly concomitant to liver injuries. Blood glucose differed widely and independently of the extent of pancreatic injury, namely from 5.05 to 24.81 mmol/L. Based on the limited number of those affected, the authors could not conclude that pancreatic injuries had a relevant impact on initial blood glucose in this patient population.

### 3.2. Liver Injuries

We found 456 patients with liver injuries; they were classified as 80 patients with AAST I, 125 patients with AAST II, 176 patients with AAST III, 64 patients with AAST IV, and 11 patients with AAST V. Overall distribution of median blood glucose levels among groups AAST I to V, differed highly significantly (Figure 2). Initial blood glucose levels increased highly significantly from 6.58 mmol/L (4.50–24.64 mmol/L) in patients with AAST I injuries to 8.60 mmol/L (5.55–27.19 mmol/L) in patients with AAST IV injuries (*p* < 0.0001). In patients with AAST V injuries, median initial blood glucose dropped to 5.77 mmol/L (2.33–7.83 mmol/L), which was significantly lower than in AAST I (*p* = 0.01) and IV injuries (*p* < 0.0001) (Figure 2a).

Exclusion of patients with diabetes mellitus (*n* = 6) resulted in very similar results (Figure 2b). Exclusion of patients with severe co-injuries (AIS 4–6, *n* = 200) (Figure 2c) and, additionally, patients with diabetes mellitus (Figure 2d) demonstrated more clearly the effects of injury severity on blood glucose levels (Table 3).

Of 11 patients with AAST V injuries, only two deceased: one due to uncontrollable bleeding of the liver and partial avulsion of the inferior vena cava, one due to uncontrollable bleeding of the liver and multiple fractures to the pelvis. Of the nine surviving patients with AAST V injury, only one required glucose administration to maintain physiological blood glucose levels during the days subsequent to admission. This was a 13-year-old child, who also needed repeated administration of coagulation factors. In all other patients, blood glucose levels improved over time without additional glucose administration. In patients with initially low or very low (hypoglycemic) blood glucose levels, blood glucose levels recovered spontaneously to slightly increased blood glucose levels following severe trauma. Apparently, the residual hepatic tissue was able to cover physiologic needs, but needed several hours to adapt. Nevertheless, the relatively low blood glucose levels were of high diagnostic importance.

### 3.3. Spleen Injuries

We found 375 patients with spleen injuries; they were graded as 62 patients with AAST I, 83 patients with AAST II, 142 patients with AAST III, 59 patients with AAST IV, and 29 patients with AAST V injuries. Overall distribution of median blood glucose levels among groups AAST I to V differed highly significantly (Figure 3). Initial blood glucose levels rose highly significantly from 7.13 mmol/L (4.83–24.81 mmol/L) in patients with AAST I to 10.71 mmol/L (5.61–18.48 mmol/L) in patients with AAST V injuries. Initial blood glucose in patients with AAST V injuries rose comparably; there was no drop (Figure 3a). Exclusion of patients with diabetes mellitus (*n* = 6) resulted in very similar results (Figure 3b). Exclusion of patients with severe co-injuries (AIS 4–6, *n* = 163) (Figure 3c) and, additionally, patients with diabetes mellitus (Figure 3d) eliminated outliers and demonstrated more clearly that initial blood glucose rose continuously with rising severity of injury and blood loss. Blood glucose in patients with AAST IV and AAST V injuries of the spleen differed highly significantly from that of patients with AAST I injuries (Figure 3c,d, Table 3).

### 3.4. Kidney Injuries

Kidney injuries were far more seldom. Overall, 152 patients were analyzed; they were graded as 21 patients with AAST I, 36 patients with AAST II, 41 patients with AAST III, 48 patients with AAST IV, and six patients with AAST V injuries. Overall distribution of median blood glucose levels among groups AAST I to V differed significantly. Initial blood glucose levels rose highly significantly from 6.49 mmol/L (5.27–11.16 mmol/L) in patients with AAST I to 9.60 mmol/L (8.88–18.48 mmol/L) in patients with AAST V injuries. There was no drop in median initial blood glucose in patients with AAST V injuries (Figure 4a). Exclusion of patients with diabetes mellitus (*n* = 3) resulted in similar results (Figure 4b). Exclusion of patients with severe co-injuries (AIS 4–6, *n* = 73) (Figure 4c) and patients with diabetes mellitus (Figure 4d) demonstrated more clearly that initial blood glucose rose continuously with rising severity of injury and blood loss. Blood glucose in patients with AAST IV and AAST V injuries of the kidney differed highly significantly from that of patients with AAST I injuries (Figure 4c,d, Table 3).

## 4. Discussion

The results of this single-center trial demonstrate that stress hyperglycemia develops parallel to injury severity in patients with blunt spleen or kidney trauma. While consistent results were obtained for blood glucose levels in blunt liver injuries up to AAST IV, stress hyperglycemia was completely absent in the most severe cases of liver injury (AAST V), which is demonstrated in this trial for the first time.

Both a pre-diagnosed diabetes mellitus and severe co-injuries (AIS 4–6) resulted in outliers of high blood glucose levels in all grades of injury severity for all three analyzed organ injuries (Figure 2, Figure 3 and Figure 4). However, they had no impact on the main results. Therefore, we did not exclude them from analysis as we felt they realistically represent the expectable extent of clinical settings. Stress hyperglycemia is a common finding in any grade of trauma severity and increases with rising severity [45,46,47,48,49]. This trial demonstrates that also in blunt, single or combined injuries to parenchymatous organs, such as liver, spleen or kidney, stress hyperglycemia accrues with the increasing extent of tissue damage and hemorrhage, thereby proving the thesis of evolution of stress hyperglycemia following trauma as set forth in the Introduction. The finding that severe co-injuries (AIS 4–6) and a pre-diagnosed diabetes mellitus increase the extent of stress hyperglycemia in some patients is also in accordance with the current literature: Rau as well as Kerby and colleagues demonstrated that stress-induced hyperglycemia exists parallel with diabetic hyperglycemia in trauma patients, but is more associated with mortality [50,51,52].

In the literature, the kidneys are also seen as a relevant source of blood glucose production (up to 40% of circulating blood glucose) following epinephrine stimulation [53,54]. Although the body’s own catecholamines should be relevantly increased during severe hemorrhage, which is a criterion for AASTV liver injury, blood glucose dropped in these patients and was not increased by renal glucose delivery. Possible attempts to explain this phenomenon include insufficient renal perfusion during hemorrhagic shock as well as mere lack of time or substances (glutamine, glycerol, lactate) for sufficient renal glucose production. In particular, circulating glutamine is mainly provided by the liver and, therefore, in the event of AAST V injury of the liver, probably lacking.

This study has several limitations. The study did not differentiate between adults, children and diabetics. We did not find significant differences between adults and children, probably due to the low median age of analyzed patients. Pre-hospital vital parameters were not sufficiently documented within the digitalized local hospital information system during the initial decade of the study phase. Therefore, detailed calculation of trauma scores was not possible. Further limitations of this study are the single-center and register design as well as the small patient number in certain subgroups. Nevertheless, this large database of patients with blunt hepatic, splenic, and renal injuries sufficiently demonstrated significant results, especially regarding median blood glucose drop in most severe liver injuries (AAST V). Furthermore, we did not analyze lactatemia due to unreliability of initial laboratory results on hospital admission [10]. Lactatemia is influenced by circulation, production, metabolism and volume resuscitation as well as administration (Ringer’s lactate), if applicable.

Therefore, initial laboratory results may fundamentally vary, not always reflecting severity of injury and hemorrhage. In contrast, blood glucose levels on hospital admission seem to be independent of pre-hospital fluid and volume administration and a reliable, additional diagnostic tool for rapid evaluation of trauma patients including patients with blunt abdominal trauma without collinearity to vital or laboratory parameters commonly used for initial trauma assessment [10]. In the case of hyperglycemia, the extent correlates with injury severity. However, in the case of a critically injured patient presenting with low blood glucose levels, it should be clarified whether—in addition to most severe liver injuries—anti-diabetic drug overdose, alcohol abuse, severe hypothermia or any other reason for limited liver function could be causative [37,55,56]. Furthermore, to distinguish between severe AAST IV and AAST V injuries of the liver—e.g., for scientific purposes—which can be challenging for surgeon or radiologists, glucose measurement may be helpful.

Future prospective studies or registers should confirm the findings of this trial. More studies on the evolution of stress hyperglycemia, its detailed effects on outcome of patients with sepsis or trauma and potential treatment options to limit the well-described negative outcome association are necessary.

## 5. Conclusions

Absence of stress hyperglycemia on hospital admission could be a sign of most severe liver injury (AAST V). Blood glucose should be considered as an additional diagnostic criterion for grading liver injury.

## Figures and Tables

**Figure 1 diagnostics-11-01667-f001:**
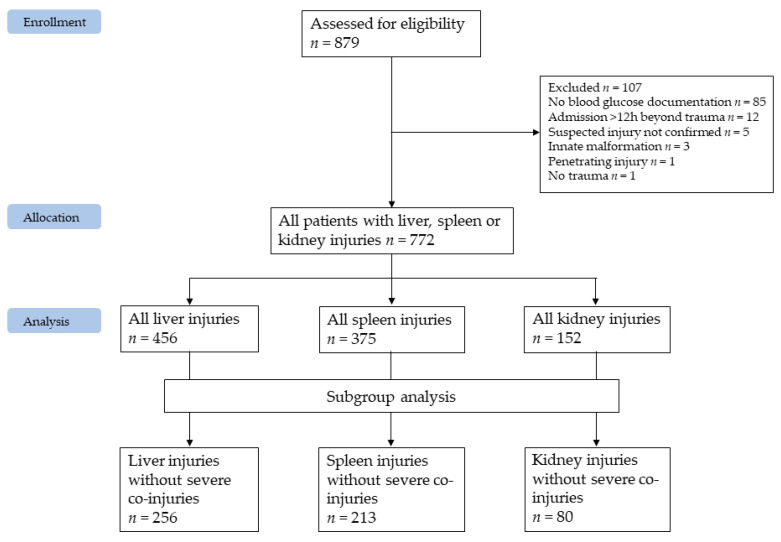
CONSORT flow diagram of patient population.

**Figure 2 diagnostics-11-01667-f002:**
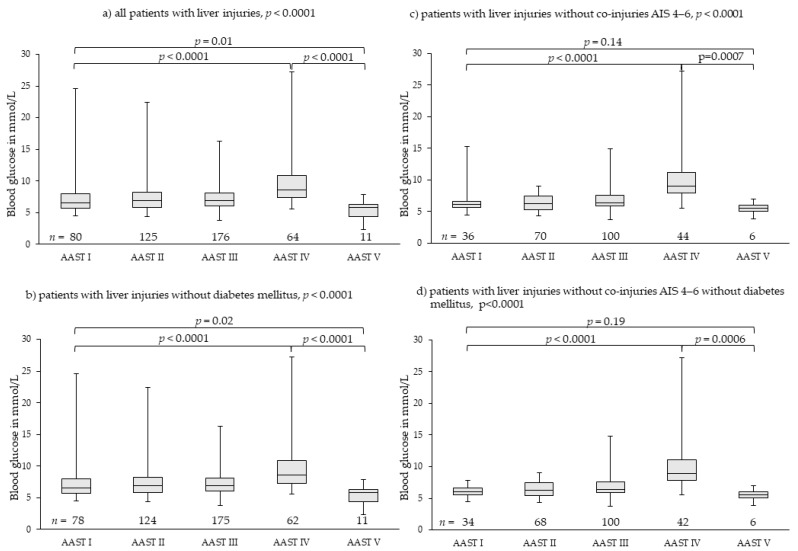
Rising blood glucose levels in parallel to severity of liver injury with lowest blood glucose levels in patients with most severe liver injury (AAST V). Outliers of high blood glucose levels in less severe injuries (AAST I–III) were found in patients with diabetes mellitus disease or severe co-injuries (AIS 4–6) (**a**–**c**). In patients without severe co-injuries and without diabetes mellitus (**d**), the effect of injury severity and hemorrhage on blood glucose levels was demonstrated, meaning the greater the severity of injury and the hemorrhage extent, the higher the stress hyperglycemia. In patients with most severe liver injury (AAST V), stress hyperglycemia is absent. AAST: American Association for the Surgery of Trauma [27,28]; AIS: Abbreviated Injury Scale [43].

**Figure 3 diagnostics-11-01667-f003:**
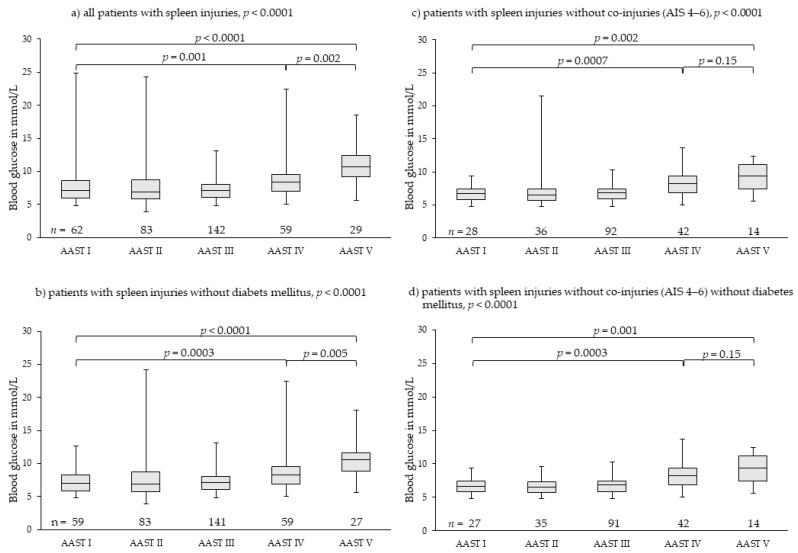
Rising blood glucose levels parallel to injury severity and hemorrhage in splenic injuries. Outliers of very high blood glucose levels in less severe injuries (AAST I-III) were found in patients with diabetes mellitus disease or severe co-injuries (AIS 4–6) (**a**–**c**). In patients without severe co-injuries and without diabetes mellitus (**d**), the effect of injury severity and hemorrhage on blood glucose levels was demonstrated more clearly, meaning the greater the severity of injury and the hemorrhage extent, the higher the stress hyperglycemia including most severe splenic injuries (AAST V). AAST: American Association for the Surgery of Trauma [27,28]; AIS: Abbreviated Injury Scale [43].

**Figure 4 diagnostics-11-01667-f004:**
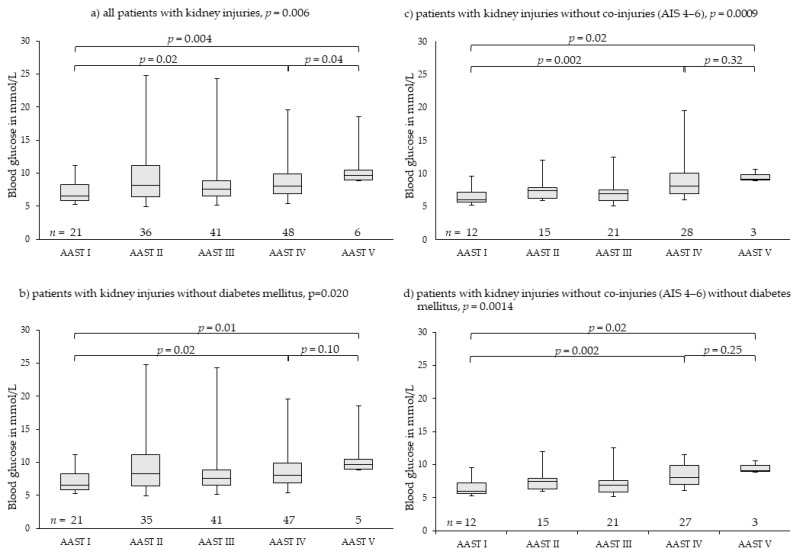
Rising blood glucose levels parallel to injury severity in renal injuries. Outliers of very high blood glucose levels in less severe injuries (AAST I-III) were found in patients with diabetes mellitus disease or severe co-injuries (AIS 4–6) (**a**–**c**). In patients without severe co-injuries and without diabetes mellitus (**d**), the effect of injury severity and hemorrhage on blood glucose levels was demonstrated more clearly, meaning the greater the severity of injury and the hemorrhage extent, the higher the stress hyperglycemia including most severe renal injuries (AAST V). AAST: American Association for the Surgery of Trauma [27,28]; AIS: Abbreviated Injury Scale [43].

**Table 1 diagnostics-11-01667-t001:** Organ Injury Scale (OIS) of the American Association for Surgery of Trauma (AAST): 2018 revision.

AAST	AIS		Liver	Spleen	Kidney
I	2	Hematoma	Subcapsular, <10% surface area	Subcapsular, <10% surface area	Subcapsular hematoma and/or parenchymal contusion without laceration
		Laceration	Capsular tear, <1% parenchymal depth	Capsular tear, <1% parenchymal depth	
II	2	Hematoma	Subcapsular, 10–50% surface area, intra-parenchymal, <10 cm in diameter	Subcapsular, 10–50% surface area Intra-parenchymal, <5 cm in diameter	Perirenal hematoma confined to Gerota fascia
		Laceration	Capsular tear, 1–3 cm parenchymal depth, <10 cm length	1–3 cm parenchymal depth	Renal parenchymal laceration ≤1 cm depth without urinary extravasation
III	3	Hematoma	Subcapsular, >50% surface area of ruptured subcapsular or parenchymal hematoma; intraparenchymal >10 cm	Subcapsular, >50% surface area Ruptured subcapsular or parenchymal hematoma ≥5 cm	
		Laceration	Capsular tear, >3 cm parenchymal depth	>3 cm parenchymal depth or involving trabecular vessels	Renal parenchymal laceration >1 cm depth without collecting system rupture or urinary extravasation
		Vascular	Vascular injury with active bleeding contained within liver parenchyma		Any injury in the presence of a kidney vascular injury or active bleeding contained within Gerota fascia
IV	4	Laceration	Parenchymal disruption involving 25–75% of hepatic lobe or 1–3 segments	Parenchymal laceration involving segmental or hilar vessels producing >25% devascularization	Parenchymal laceration extending into urinary collecting system with urinary extravasation Renal pelvis laceration and/or complete ureteropelvic disruption
		Vascular	Vascular injury with active bleeding breaching the liver parenchyma into the peritoneum	Any injury in the presence of a splenic vascular injury or active bleeding confined within splenic capsule	Segmental renal vein or artery injury Active bleeding beyond Gerota fascia into the retroperitoneum or peritoneum Segmental or complete kidney infarction(s) due to vessel thrombosis without active bleeding
V	5	Laceration	Parenchymal disruption involving >75% of hepatic lobe	Shattered spleen	Shattered kidney with loss of identifiable parenchymal renal anatomy
		Vascular	Juxtavenous hepatic injuries; i.e., retrohepatic vena cava/central major hepatic veins	Any injury in the presence of splenic vascular injury with active bleeding extending beyond the spleen into the peritoneum	Main renal artery or vein laceration or avulsion of hilum Devascularized kidney with active bleeding

Additional points: Advance one grade for multiple injuries up to Grade III. Vascular injury (i.e., pseudoaneurysm or AV fistula) appears as a focal collection of vascular contrast that decreases in attenuation on delayed images. *Active bleeding*: focal or diffuse collection of vascular contrast that increases in size or attenuation in a delayed phase. Details from [40,41].

**Table 2 diagnostics-11-01667-t002:** Study population characteristics, *n* = 772.

Characteristics	Median/Number	Range/Percent
Age	29	1–89
female/male	238/534	30.8/69.2
Body mass index (BMI)	23.0	13.1–43.2
Deceased	26	3.4
Hospital length of stay (d)	13	0–112
Blood glucose (mmol/L) ^1^	7.16	2.33–27.19 ^1^
ISS ^2^	21	1–75
Accident mechanism		
Accidents with skis, sleigh, or snowboard	301	39.0
Accident with motor vehicle	148	19.2
Accident with moped or motorbike	81	10.5
Accident with bike, scooter, skateboard	71	9.2
Fall > 3 m	67	8.7
Occupational injury	28	3.7
Unknown mechanism	19	2.5
Accident with large animal	18	2.3
Accident as pedestrian	14	1.8
Fall < 3 m	11	1.4
Accident with flex wing, parachute or similar	6	0.8
Fall down stairs	5	0.6
Brawl	3	0.4
Co-morbidities		
Arterial hypertension	36	4.7
Other disease	35	4.5
Psychiatric disease	25	3.2
Allergy/atopic dermatitis	22	2.8
Neurologic disease	18	2.3
Cardiac insufficiency and other cardiac diseases	16	2.1
Diabetes mellitus	13	1.7
Alcohol abuse	14	1.8
Adiposity	12	1.6
Endocrinologic disease	11	1.4
Malign disease	10	1.3
Chronic obstructive pulmonary disease	8	1.0
Renal insufficiency	8	1.0
Autoimmune disease	8	1.0
Metabolic disease	6	0.8
Coronary heart disease	5	0.6
Bronchial asthma	5	0.6
Laboratory findings ^1^		
Hemoglobin mg/dL	121	34–187
Thrombocytes G/L	198	14–835
Leukocytes G/L	13.1	1.5–39.1
Lactate mg/dL	15.4	4–176
pH	7.367	6.882–7.642
Bicarbonate standardized mmol/L	22.1	8.3–27.0
Base excess mmol/L	−2.0	−15.6–11.0
Potassium mmol/L	3.8	2.32–7.62
Anion gap mmol/L	21.8	8.3–27.0

^1^ Measured within first hour following admission; ^2^ Injury Severity Score [44].

**Table 3 diagnostics-11-01667-t003:** Blood glucose levels in mmol/L for liver, spleen and kidney injuries as function of injury severity with and without co-injuries AIS 4–6 and diabetes mellitus.

	AAST I	AAST II	AAST III	AAST IV	AAST V	*p*	AAST I	AAST II	AAST III	AAST IV	AAST V	*p*
Liver	**All patients with liver injuries**						**Patients with liver injuries without co-injuries AIS 4–6**					
Median	6.58	6.88	6.94	8.60	5.77	<0.0001	6.16	6.22	6.41	8.96	5.58	<0.0001
Min-max	4.50–24.64	4.33–22.42	3.77–16.32	5.55–27.19	2.33–7.83		4.50–15.26	4.33–9.05	3.77–14.87	5.55–27.19	3.83–6.94	
IQR	5.70–7.95	5.77–8.21	6.05–8.12	7.31–10.91	4.36–6.33		5.63–6.63	5.34–7.44	5.87–7.56	7.99–11.20	5.01–6.06	
CI 95%	6.90–8.39	7.05–8.04	7.11–7.71	8.82–10.80	4.32–6.29		5.90–7.12	6.15–6.71	6.54–7.19	9.07–11.75	4.71–6.28	
	**Patients with liver injuries without diabetes mellitus**						**Patients with liver injuries without co-injuries AIS 4–6 without diabetes mellitus**					
Median	6.52	6.80	6.94	8.52	5.77	<0.0001	6.08	6.22	6.41	8.91	5.58	<0.0001
Min-max	4.50–24.64	4.33–22.42	3.77–16.32	5.55–27.19	2.33–7.83		4.50–7.77	4.33–9.05	3.77–14.87	5.55–27.19	3.83–6.94	
IQR	5.67–7.83	5.77–8.21	6.05–8.13	7.17–10.75	4.36–6.33		5.58–6.59	5.37–7.44	5.87–7.56	7.88–11.07	5.01–6.06	
CI 95%	6.77–8.24	7.04–8.04	7.11–7.71	8.59–10.36	4.42–6.19		5.85–6.40	6.14–6.70	6.54–7.19	8.75–11.15	4.71–6.28	
Spleen	**All patients with spleen injuries**						**Patients with spleen injuries without co-injuries AIS 4–6**					
Median	7.13	6.88	7.08	8.32	10.71	<0.0001	6.72	6.55	6.88	8.21	9.41	<0.0001
Min-max	4.83–24.81	3.88–24.25	4.83–13.10	5.05–22.42	5.61–18.48		4.83–9.43	4.83–21.48	4.83–10.32	5.05–13.65	5.61–12.43	
IQR	5.90–8.57	5.80–8.69	6.10–8.03	6.94–9.55	9.16–12.43		5.83–7.49	5.72–7.38	5.88–7.44	6.91–9.42	7.42–11.17	
CI 95%	6.98–8.39	7.17–8.58	6.96–7.46	8.23–9.92	9.74–12.16		6.34–7.28	6.18–7.97	6.58–7.07	7.96–8.61	8.09–10.40	
	**Patients with spleen injuries without diabetes mellitus**						**Patients with spleen injuries without co-injuries AIS AIS 4–6without diabetes mellitus**					
Median	6.99	6.88	7.10	8.32	10.60	<0.0001	6.66	6.49	6.88	8.21	9.41	<0.0001
Min-max	4.83–12.71	3.88–24.25	4.83–13.10	5.05–22.42	5.61–18.09		4.83–9.43	4.83–9.60	4.83–10.32	5.05–13.65	5.61–12.43	
IQR	5.88–8.30	5.80–8.69	6.10–8.05	6.94–9.55	8.91–11.58		5.83–7.38	5.72–7.30	5.88–7.44	6.91–9.42	7.42–11.17	
CI 95%	6.88–7.41	7.13–8.39	6.96–7.46	8.23–9.29	9.74–12.16		6.27–7.19	6.23–7.10	6.58–7.07	7.72–8.85	8.09–10.40	
Kidney	**All patients with kidney injuries**						**Patients with kidney injuries without co-injuries AIS 4–6**					
Median	6.49	8.21	7.60	8.02	9.60	0.006	6.02	7.44	6.94	8.10	9.10	0.0009
Min-max	5.27–11.16	4.94–24.81	5.11–24.25	5.44–19.59	8.88–18.48		5.27–9.60	5.94–11.99	5.11–12.54	6.05–19.59	8.88–10.60	
IQR	5.88–8.27	6.44–11.16	6.55–8.88	6.94–9.84	8.94–10.48		5.66–7.21	6.33–7.88	5.88–7.60	6.98–10.05	8.99–9.85	
CI 95%	6.52–7.85	7.35–11.27	7.26–9.24	7.87–9.36	8.29–13.73		5.87–7.30	6.67–8.16	6.45–7.81	7.82–9.80	8.66–10.39	
	**Patients with kidney injuries without diabetes mellitus**						**Patients with kidney injuries without co-injuries AIS 4–6without diabetes mellitus**					
Median	6.49	7.94	7.60	7.99	9.10	0.02	6.02	7.44	6.94	8.10	9.10	0.001
Min-max	5.27–11.16	4.94–22.42	5.11–24.25	5.44–13.62	8.88–10.60		5.27–9.60	5.94–11.99	5.11–12.54	6.05–11.54	8.88–10.60	
IQR	5.88–8.27	6.44–11.04	6.55–8.88	6.94–9.60	8.88–10.10		5.66–7.21	6.33–7.88	5.88–7.60	6.97–9.91	8.99–9.85	
CI 95%	6.52–7.85	8.14–10.62	7.26–9.24	7.77–8.99	7.06–11.97		5.87–7.30	6.67–8.16	6.45–7.81	7.76–9.06	8.66–10.39	

CI: Confidence interval; range: minimum and maximum; IQR: interquartile range.

## Data Availability

All data that support the findings of this study are available from the corresponding author upon reasonable request.

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
