# Peer review of "Absence of Stress Hyperglycemia Indicates the Most Severe Form of Blunt Liver Trauma"

_diagnostics, 2021, doi:10.3390/diagnostics11091667_

Round 1
Reviewer 1 Report
A good paper about the usefulness of blood glucose levels as an additional diagnostic criterion for liver injury with a high injury score scale value (V).
Author Response
Thank you very much for your compliments and favorable review. We hope our answers and amendments to the other reviewers are adequate.
Reviewer 2 Report
The authors investigated absence of stress hyperglycemia in the most severe form of blunt liver trauma. They concluded that absence of stress hyperglycemia upon hospital admission could be a sign of most severe liver injury and that blood glucose should be considered as an additional diagnostic criterion for grading liver injury.
This is interesting and well written study with adequate and reproducible methodology. I have several suggestions for improvement:
- Title – Avoid abbreviations in the title of the manuscript
- The authors used several abbreviations in abstract but they did not explain the meaning of the abbreviations
- Introduction is poor. More details about glucose metabolism in stress reactions should be provided.
- Please indicate primary and secondary outcomes of the study in methodology.
- Table 1. – Study population – Please provide more basic data regarding the patients (BMI, comorbidities…)
- Please provide a Tables with AAST and ISS (it would be easier to the readers to follow the data).
- Discussion is poorly designed. Discussion section needs to be re-written/re-arranged. Do not present review of literature in this section. Do not discuss your findings piecemeal. Focus on results from main objectives of the study. Write in four sequential paragraphs (without headings); (i) summary (not data) of findings from present study; (ii) logical and coherent comparison with existing literature with focus of comparison on main objective(s); (iii) limitations of the study; and (iv) Implications for practice/policy/research with a concluding statement.
Author Response
The authors would like to thank you for the detailed reading and your critical and helpful comments. We marked the changes in the manuscript in yellow.
Title – Avoid abbreviations in the title of the manuscript
We removed the abbreviation in the title.
The authors used several abbreviations in abstract but they did not explain the meaning of the abbreviations
We added “according to American Association for the Surgery of Trauma” in the abstract.
Introduction is poor. More details about glucose metabolism in stress reactions should be provided.
We added a paragraph with more detailed pathophysiological background information on emergence of stress hyperglycemia following trauma according to current literature. The introduction now reads as follows:
This stress hyperglycemia is caused by neuroendocrine, inflammatory and metabolic responses to trauma-associated stressors. Pain, anxiety and psychogenic stress may lead to activation of the sympathetic nervous system and the hypothalamic-pituitary axis, increasing circulating catecholamines. A traumatic brain injury may even lead to a sympathetic storm [8,9], thus further amplifying this neuroendocrine response. Another very strong trigger for a stress response such as stress hyperglycemia is hemorrhage [10], which is mediated by baroreceptors and multifactorially released cytokines. [11] Furthermore, tissue hypoxemia and injury, endothelial damage and complement activation [12,13] kick off an avalanche of immune-stimulating and -modulating chemokines, interleukins, [14-16] damage-associated molecular patterns (DAMPS) and multiple cell line activations. [13,17] Circulating and locally secreted tumor necrosis factor (TNF)α [18,19] and catecholamines [20] lead to hepatic insulin resistance, which annuls glucose homeostasis. By triggering rampant hepatic gluconeogenesis and glycogenolysis both processes are decoupled from physiological feedback mechanisms [21-23] and, additionally, are fed by free fatty acids released by catecholamines from fat tissue, [24,25] but also from lactate and alanine of muscle tissue. [25-27]
TNFα-associated hepatic insulin resistance is probably mediated by TNF receptors by activating a kinase (c-Jun-NH2-terminal kinase), which in turn catalyzes serine phosphorylation of insulin receptor substrate 1 (IRS-1). This blocks the phosphatidylinositol 3-kinase (PI3K) and protein kinase B (AKT) pathway of intracellular metabolic effects of insulin. [28] Catecholamine (epinephrine)-mediated insulin resistance is conveyed via 3’–5’-cyclic adenosine monophosphate (cyclic AMP or cAMP) [29], activating protein kinase A (PKA). [30] On the one hand, phosphorylase kinase is activated, itself stimulating glycogen phosphorylase and therefore leading to a disintegration of glycogen stores and release of glucose. [31,32] On the other hand, PKA inhibits in parallel substrate flux through phosphofructokinase-1 (PFK-1) (rate-limiting enzyme in glycolysis), thereby enhancing the delivery of glucose by the liver. [33,34] Moreover, peripheral insulin resistance – mainly of the muscles and probably by comparable mechanisms – limits glucose uptake and metabolism. [21,35]
Probably, the pathophysiological purpose of all these varying responses to trauma is …
Please indicate primary and secondary outcomes of the study in methodology.
Thank you for this important annotation. We added the following paragraph to the method section.
Primary outcome parameter was the admission blood glucose level depending on severity of injury (AAST I to V) to liver, spleen, and kidney. Secondary outcome parameters were admission blood glucose levels additionally depending on the existence of diabetes mellitus and severe co-injuries (AIS 4-6)
Table 1. – Study population – Please provide more basic data regarding the patients (BMI, comorbidities…)
Due to the very young median age (29 years as documented in former table 1, most patients were slim and had no co-morbidities. However, we added the following information to table 2: BMI, co-morbidities, and mechanisms of injury.
Please provide a Tables with AAST and ISS (it would be easier to the readers to follow the data).
We added a table with grading of organ injuries according to AAST and AIS (because ISS grading of one organ is in our humble opinion insufficient) according to current literature (Kozar RA et al. Organ injury scaling 2018 update: Spleen, liver, and kidney. J Trauma Acute Care Surg. 2018; 85:1119-22; Morell-Hofert D et al. Validation of the revised 2018 AAST-OIS classification and the CT severity index for prediction of operative management and survival in patients with blunt spleen and liver injuries. Eur Radiology 2020; 30:6570-81)
Discussion is poorly designed. Discussion section needs to be re-written/re-arranged. Do not present review of literature in this section. Do not discuss your findings piecemeal. Focus on results from main objectives of the study. Write in four sequential paragraphs (without headings); (i) summary (not data) of findings from present study; (ii) logical and coherent comparison with existing literature with focus of comparison on main objective(s); (iii) limitations of the study; and (iv) Implications for practice/policy/research with a concluding statement.
We re-arranged and re-wrote the discussion section according your input requirement. The discussion section now reads as follows:
The results of this single-center trial demonstrate that stress hyperglycemia develops parallel to injury severity in patients with blunt spleen or kidney trauma. While consistent results were obtained for blood glucose levels in blunt liver injuries up to AAST IV, stress hyperglycemia was completely absent in the most severe cases of liver injury (AAST V), which is demonstrated in this trial for the first time.
Both a prediagnosed diabetes mellitus and severe co-injuries (AIS4-6) resulted in outliers of high blood glucose levels in all grades of injury severity for all three analyzed organ injuries (Figures 2-4). However, they had no impact on the main results. Therefore, we did not exclude them from analysis as we felt they realistically represent the expectable extent of clinical settings. Stress hyperglycemia is a common finding in any grade of trauma severity and increases with rising severity. [45-49] This trial demonstrates that also in blunt, single or combined injuries to parenchymatous organs such as liver, spleen or kidney stress hyperglycemia accrues with the increasing extent of tissue damage and hemorrhage, thereby proving the thesis of evolution of stress hyperglycemia following trauma as set forth in the Introduction. The finding that severe co-injuries (AIS 4-6) and a prediagnosed diabetes mellitus increase the extent of stress hyperglycemia in some patients is also in accordance with current literature: Rau as well as Kerby and colleagues demonstrated that stress-induced hyperglycemia exists parallel with diabetic hyperglycemia in trauma patients, but is more associated with mortality. [50-52]
In the literature, the kidneys are also seen as a relevant source of blood glucose production (up to 40% of circulating blood glucose) following epinephrine stimulation. [53,54] Although the body’s own catecholamines should be relevantly increased during severe hemorrhage, which is a criterion for AASTV liver injury, blood glucose dropped in these patients and was not increased by renal glucose delivery. Possible attempts to explain this phenomenon include insufficient renal perfusion during hemorrhagic shock as well as mere lack of time or substances (glutamine, glycerol, lactate) for sufficient renal glucose production. In particular, circulating glutamine is mainly provided by the liver and therefore, in the event of AAST V injury of the liver, probably lacking.
This study has several limitations. The study did not differentiate between adults, children and diabetics. We did not find significant differences between adults and children, probably due to the low median age of analyzed patients. Pre-hospital vital parameters were not sufficiently documented within the digitalized local hospital information system during the initial decade of the study phase. Therefore, detailed calculation of trauma scores was not possible. Further limitations of this study are the single-center and register design as well as the small patient number in certain subgroups. Nevertheless, this large database of patients with blunt hepatic, splenic, and renal injuries sufficiently demonstrated significant results, especially regarding median blood glucose drop in most severe liver injuries (AAST V). Furthermore, we did not analyze lactatemia due to unreliability of initial laboratory results on hospital admission. [10] Lactatemia is influenced by circulation, production, metabolism and volume resuscitation as well as administration (Ringer’s lactate), if applicable. Therefore, initial laboratory results may fundamentally vary, not always reflecting severity of injury and hemorrhage. In contrast, blood glucose levels on hospital admission seem to be independent of pre-hospital fluid and volume administration and a reliable, additional diagnostic tool for trauma patients without collinearity to vital or laboratory parameters commonly used for initial trauma assessment. [10] However, if critically injured patients present with low blood glucose levels, it should be clarified whether – in addition to most severe liver injuries - anti-diabetic drug overdose, alcohol abuse, severe hypothermia or any other reason for limited liver function could be causative. [37,55,56]
We left the conclusion as in the initial version of manuscript.
Reviewer 3 Report
Major Comments:
- The introduction is insufficient to a new reader to understand the background so please expand.
- The differences in blood glucose levels (absolute values) in different patients under different conditions for each injury type will be better understood if they were represented in a table form in addition to the text format.
- The authors use extremely long sentences in the results sections, such lines should be rephrased and make it much more understanding without loosing the interest of the reader.
For eg: Line 174 ''Excluding patients with severe co-injuries (AIS 4-6, n=73) (Figure 4c) and, additionally, patients with diabetes mellitus (Figure 4d) demonstrated – due to less exceptions and outliers - more clearly that initial blood glucose rose continuously with rising injury severity of the kidneys and blood losses.''
Other examples: Lines 154,
Minor comments:
Please check the entire manuscripts for spelling corrections.
Author Response
The introduction is insufficient to a new reader to understand the background so please expand.
We extensively expanded the introduction section, which now reads as follows:
Stress hyperglycemia is common in trauma patients, and critical illness on hospital admission and is often associated with poor outcome. [1-7] This stress hyperglycemia is caused by neuroendocrine, inflammatory and metabolic responses to trauma-associated stressors. Pain, anxiety and psychogenic stress may lead to activation of the sympathetic nervous system and the hypothalamic-pituitary axis, increasing circulating catecholamines. A traumatic brain injury may even lead to a sympathetic storm [8,9], thus further amplifying this neuroendocrine response. Another very strong trigger for a stress response such as stress hyperglycemia is hemorrhage [10], which is mediated by baroreceptors and multifactorially released cytokines. [11] Furthermore, tissue hypoxemia and injury, endothelial damage and complement activation [12,13] kick off an avalanche of immune-stimulating and -modulating chemokines, interleukins, [14-16] damage-associated molecular patterns (DAMPS) and multiple cell line activations. [13,17] Circulating and locally secreted tumor necrosis factor (TNF)α [18,19] and catecholamines [20] lead to hepatic insulin resistance, which annuls glucose homeostasis. By triggering rampant hepatic gluconeogenesis and glycogenolysis both processes are decoupled from physiological feedback mechanisms [21-23] and, additionally, are fed by free fatty acids released by catecholamines from fat tissue, [24,25] but also from lactate and alanine of muscle tissue. [25-27]
TNFα-associated hepatic insulin resistance is probably mediated by TNF receptors by activating a kinase (c-Jun-NH2-terminal kinase), which in turn catalyzes serine phosphorylation of insulin receptor substrate 1 (IRS-1). This blocks the phosphatidylinositol 3-kinase (PI3K) and protein kinase B (AKT) pathway of intracellular metabolic effects of insulin. [28] Catecholamine (epinephrine)-mediated insulin resistance is conveyed via 3’–5’-cyclic adenosine monophosphate (cyclic AMP or cAMP) [29], activating protein kinase A (PKA). [30] On the one hand, phosphorylase kinase is activated, itself stimulating glycogen phosphorylase and therefore leading to a disintegration of glycogen stores and release of glucose. [31,32] On the other hand, PKA inhibits in parallel substrate flux through phosphofructokinase-1 (PFK-1) (rate-limiting enzyme in glycolysis), thereby enhancing the delivery of glucose by the liver. [33,34] Moreover, peripheral insulin resistance – mainly of the muscles and probably by comparable mechanisms – limits glucose uptake and metabolism. [21,35]
Probably, the pathophysiological purpose of all these varying responses to trauma is to shift energy substrates to vital organs, initiate immune defense, and repair mechanisms [20-22,25,26,36], which can therefore be seen as survival responses. Consequently, according to the literature, hypoglycemia is a rare finding in trauma patients at hospital admission and is triggered mainly by non-traumatic causes such as anti-diabetic drug overdose, alcohol intoxication or chronic liver disease. [37,38] To date, there has been no consistency in defining blood glucose levels, for which reason this trial refrains from specifying any threshold values for stress hyperglycemia.
Injury severity of parenchymatous organs like liver, kidney or spleen is often radiologically categorized according to the classification of the American Association for the Surgery of Trauma, AAST. [39,40, Table 1] In doing so, the extent of lacerations, contusions or hematomas must be exactly measured. However, specifying proportions of parenchymal disruption of one or both hepatic lobes to distinguish between liver injury AAST IV and V may be very challenging. Active bleeding is included in AAST≥III injuries of the liver and kidney and AAST≥IV injuries of the spleen. [39,40] AAST V injuries of the liver are furthermore defined as major juxtahepatic venous injury (vena cava, central major hepatic veins) or lacerations resulting in parenchymal disruption of more than 75% of one hepatic lobe. [39,40] Assuming the liver is the primary origin of stress hyperglycemia, it seems clear that such destructive liver injuries, partially even resulting in devascularization (inflow or outflow), can impede hepatic glucose liberation provoked by trauma or hemorrhage.
The differences in blood glucose levels (absolute values) in different patients under different conditions for each injury type will be better understood if they were represented in a table form in addition to the text format.
As you wished, we added a table with results in absolute values.
The authors use extremely long sentences in the results sections, such lines should be rephrased and make it much more understanding without loosing the interest of the reader.
For eg: Line 174 ''Excluding patients with severe co-injuries (AIS 4-6, n=73) (Figure 4c) and, additionally, patients with diabetes mellitus (Figure 4d) demonstrated – due to less exceptions and outliers - more clearly that initial blood glucose rose continuously with rising injury severity of the kidneys and blood losses.''
Other examples: Lines 154,
The result section was rephrased and the sentences shortened.
Minor comments:
Please check the entire manuscripts for spelling corrections.
The entire manuscript was checked by a sworn court interpreter.
Reviewer 4 Report
1. Effects of hyperglycemia are also affected by underlying diabetes. The investigators however did not look into diabetes and non-diabetes subgroup.
2. Data on baseline characteristics are very thin. Need to be provided in more details including baseline characteristics, age, sex, race, BMI, comorbidities, and detailed laboratories findings.
3. Associations with outcomes such as in-hospital mortality should be evaluated.
4. The hypothesis of the study should be explicitly stated
5. Please follow STROBE guideline by stating the type of study in the methods section. Is it a cohort, case-control, or cross-sectional study?
6. Outcomes are not defined with the methods section (or analysed appropriately in the results section)
7. Exposures were not clearly defined in the methods section
8. methods used have confused the exposures and outcomes.
9. Missing data was not described, explored or accounted for.
10. The statistical methods used were poorly described throughout.
11. P-values should be presented to 2 decimal places and effect estimates and 95%CI should be presented in all instances
12. The authors should discuss what the clinical utility of these findings could be
13. There should be more discussion of a possible mechanism
14. Explain what sort of future study might take us closer to a clinical utility
Author Response
Effects of hyperglycemia are also affected by underlying diabetes. The investigators however did not look into diabetes and non-diabetes subgroup.
The effects of (stress) hyperglycemia are not topic of this manuscript. In this manuscript stress hyperglycemia was analyzed as a diagnostic criterion upon hospital admission. Outcome is not a study parameter, because outcome of this study population has already been published in Fodor M, Primavesi F, Morell-Hofert D, Kranebitter V, Palaver A, Braunwarth E, et al. Non-operative management of blunt hepatic and splenic injury: a time-trend and outcome analysis over a period of 17 years. World J Emerg Surg. 2019;14:29 and cited in the manuscript.
However, we did look into diabetes, by analyzing patients with and without diabetes mellitus (figure 2, 3, 4 each b and d, table 3), thereby shedding light on this topic. In sum, 13 of 772 included patients (1.7%) had a diagnosed diabetes mellitus. Therefore, this population was too small to be sufficiently analyzed in comparison to patients without diabetes mellitus. In this trial, patients with diabetes mellitus tended to have overshooting hyperglycemia in comparison to patients without diabetes mellitus and comparable injury severity. Please have a look at figures 2-4 and table 3.
Data on baseline characteristics are very thin. Need to be provided in more details including baseline characteristics, age, sex, race, BMI, comorbidities, and detailed laboratories findings.
Median age and sex has already been provided in table 2. We added the BMI and comorbidities, and laboratory findings. Due to the young median age (29 years), comorbidities were rare and their influence on laboratory values were low, maybe even insignificant. Races are not documented in our local hospital information system. In the experience of the authors, the number of patients with African, Asian or South American heritage is extremely low amongst people from the catchment area (alpine surrounding, many sports accidents) of our hospital.
Associations with outcomes such as in-hospital mortality should be evaluated.
In-hospital mortality is given in table 2. In case, you wish further information, please let us know. However, as written above, this manuscript did not focus on outcome. Detailed outcome data of the biggest part of the analyzed population has already been published in Fodor M et al. Non-operative management of blunt hepatic and splenic injury: a time-trend and outcome analysis over a period of 17 years. World J Emerg Surg. 2019;14:29, which has been mentioned in the manuscript.
The hypothesis of the study should be explicitly stated
The hypothesis has been stated in the introduction section: “…, it was hypothesized that, in parallel with injury severity and extent of hemorrhage, in hepatic, renal or splenic injuries, blood glucose levels should increase and therefore lead to significant stress hyperglycemia. However, it was also hypothesized that in the case of most severe liver injuries (AAST V), defined by major devascularization (inflow or outflow) and/or parenchymal disruption of more than 75% of one hepatic lobe, hepatic gluconeogenesis and glycogenolysis may become insufficient, consequently leading to absence of stress hyperglycemia. This may be a leading diagnostic mark.”
If you need further specification, please let us know.
Please follow STROBE guideline by stating the type of study in the methods section. Is it a cohort, case-control, or cross-sectional study?
This is a descriptive observational study without any comparing group, wherefore this is neither a cohort, case-control, or cross-sectional study.
Outcomes are not defined with the methods section (or analysed appropriately in the results section)
Thank you for this valuable comment. Primary and secondary outcomes, which have been predefined, were now additionally listed in the method section. According to this, these parameters were analyzed and presented in the result section. In case you wish further analyses, it would be our pleasure, to give more detailed information according to your additional suggestions.
Exposures were not clearly defined in the methods section
We have added the accident mechanisms in table 1.
methods used have confused the exposures and outcomes.
We hope, that the list of pre-defined outcome parameters and accident mechanisms as exposures now clarify this confusion.
Missing data was not described, explored or accounted for.
Missing data is a huge problem in pre-hospital as well as in in-hospital emergency medicine. According to literature, this is a world-wide phenomenon. In most cases, involved medical staff is urgently busy with treating one or more patients thereby being obliged to disregard documentation. Therefore, as already depicted on figure 1, of the 107 excluded patients, 85 had to be excluded due to missing blood glucose documentation upon hospital admission. In addition, we have added the following paragraph in the result section:
Compared to the 772 included patients, the 107 excluded patients had a comparable median age (31 (interquartile range: 16-43) years) and gender distribution (31.8% female). Of them, 50 (46.8%) suffered from hepatic, 45 (42.1%) from spleen, and six (6.6%) patients from renal injury. In sum, 13 (12.1%) patients deceased during in-hospital stay. As depicted in Figure 1, nine patients were excluded due to malformation, missing confirmation of the suspected injury or no trauma. In sum, aside from an increased – mainly early – mortality rate, the excluded population is comparable to the included population. Therefore, it was concluded that the included population is representative and inclusion of the excluded population would not have a significant impact on the presented results.
The statistical methods used were poorly described throughout.
Due to the descriptive, observational design of the study, statistical analysis was simple, indeed. Beside descriptive analysis (median, minimum, maximum, interquartile range, frequencies) and the graphical presentation, nothing more than the mentioned Kruskal-Wallis and Mann-Whitney U test was performed. If you need further information, please let us know your detailed wishes.
P-values should be presented to 2 decimal places and effect estimates and 95%CI should be presented in all instances
We now give p values with 2 decimals where applicable.
We have added the 95% CI in Table 3 (all results), although, in our humble opinion, this is unusual for non-normally distributed data, which are commonly described by range and/or interquartile range.
The authors should discuss what the clinical utility of these findings could be
We added the following paragraph to the discussion section:
…blood glucose levels on hospital admission seem to be independent of pre-hospital fluid and volume administration and a reliable, additional diagnostic tool for rapid evaluation of trauma patients including patients with blunt abdominal trauma without collinearity to vital or laboratory parameters commonly used for initial trauma assessment. [10] In the case of hyperglycemia, the extent correlates with injury severity. However, in the case of a critically injured patient presenting with low blood glucose levels, it should be clarified whether – in addition to most severe liver injuries - anti-diabetic drug overdose, alcohol abuse, severe hypothermia or any other reason for limited liver function could be causative. [37,55,56] Furthermore, to distinct between severe AAST IV and AAST V injuries of the liver – eg for scientific purposes – which can be challenging for surgeon or radiologists, glucose measurement may be helpful.
There should be more discussion of a possible mechanism
We have extended the potential evolution of stress hyperglycemia in the trauma setting in the introduction.
Explain what sort of future study might take us closer to a clinical utility
The following sentences were added at the end of the discussion section:
Future prospective studies or registers should confirm the findings of this trial. More studies on the evolution of stress hyperglycemia, its detailed effects on outcome of patients with sepsis or trauma and potential treatment options to limit the well-described negative outcome association are necessary.
Round 2
Reviewer 2 Report
The authors adequately responded to all my comments and significantly improved the quality of manuscript. In my opinion the manuscript is acceptable for publication in present form.
Reviewer 4 Report
It appears that all comments have been appropriately responded to. I have no further comments